# Parameter Efficient Node Classification on Homophilic Graphs

**Lucas Prieto**                                               *lucas.prieto.al@gmail.com*
*Socialdatabase*
*University of Amsterdam*

**Jeroen Den Boef**                                       *jeroendenboef1998@gmail.com*
*Socialdatabase*

**Paul Groth**                                                          *p.t.groth@uva.nl*
*University of Amsterdam*

**Joran Cornelisse**                                         *joran.cornelisse@gmail.com*
*Socialdatabase*

**Reviewed on OpenReview:** *https://openreview.net/forum?id=XXXX*

## Abstract

Deep Learning on Graphs was recently made possible with the introduction of Graph Neural Networks (GNNs). GNNs use learnable diffusion processes to propagate information through the graph and improve performance on downstream tasks. However, learning this diffusion process can be expensive in terms of memory and computation. While much research has gone into making these models more expressive and able to capture more complex patterns, in practice, edges in common benchmarking datasets often encode similarity of nodes with respect to the downstream task. This property is called homophily. We argue that for these homophilic graphs, learnable diffusion processes and large receptive fields are not required to achieve competitive performance. We propose Graph Non-Parametric Diffusion (GNPD) a method that outperforms traditional GNNs using only 2 linear models and non-parameteric diffusion. Our method takes ideas from Correct & Smooth (C&S) and the Scalable Inception Graph Network (SIGN) and combines them to create a simpler model that outperforms both of them on several datasets. Our method achieves unmatched parameter efficiency, competing with models with two orders of magnitude more parameters. Additionally GNPD can also forego spectral embeddings which are the computational bottleneck of the C&S method.

## 1  Introduction

Pairwise relationships between entities arise in many settings in the real world. From people following each-other on social networks, to citations between papers (Rossi & Ahmed, 2015)(Bhattacharya & Getoor, 2007), or links between atoms in a molecule (Hu et al., 2020). These relationships can be described by a network structure which is mathematically referred to as a graph with nodes connected by edges. Since graphs can be used to reflect the structure of real world data, they are increasingly used to make predictions and perform Machine Learning (ML) in settings where relationships between entities are important predictors (Hu et al., 2020). This has led to a rising interest in finding the best way to use this graph structure to make predictions.

Harnessing the structure of data to find local patterns has been key to the success of Deep Learning (DL) tasks on images (LeCun et al., 2015)(Szegedy et al., 2014) and text (Devlin et al., 2018). However, the input

spaces for these tasks have a clear and rigid structure. Sentences are sequences of words, and images are grids of pixels. This is not the case for graphs, where each node can have a different number of neighbors which are not ordered in any particular way. Despite the challenges that come with a more complex structure, several methods have successfully implemented deep learning over graphs with Graph Neural Networks (GNNs). Common GNN methods include Graph Convolutional Networks (GCN) (Kipf & Welling, 2016), Graph Attention Networks (Veličković et al., 2018) and Message Passing GNNs (MPGNNs) (Gilmer et al., 2017). These GNNs outperformed preexisting methods on most benchmarking datasets for machine learning on graphs (Hu et al., 2020).

Despite the success of these methods, recent papers have pointed out that GNNs may have inherited some assumptions from Deep Learning, which do not hold on graphs. For example, Simplified GCN (SGCN) (Wu et al., 2019) and SIGN (Rossi et al., 2020) showed that network depth is not needed to achieve state of the art (SOTA) performance on graphs. By using shallow networks, these methods become substantially faster while requiring less memory. Similarly, the Correct & Smooth (C&S) (Huang et al., 2020) method outperformed many GNNs by using a graph agnostic Multi Layer Perceptron (MLP) combined with residual and label propagation post-processing through the graph structure. This makes it orders of magnitude more scalable than most GNN methods since the graph structure is only used once during training.

While more scalable than traditional GNNs, these methods also have shortcomings. For optimal performance, C&S requires the calculation of spectral embeddings to encode the graph structure, which can be expensive for large graphs. On the other hand, SIGN trains several models for every task, resulting in millions of parameters for most datasets (Hu et al., 2020). Finally, SIGN and SGCN were the SOTA at the time of their publication but have since been left behind by newer methods.

To address these limitations we make the following contributions: 1) Re-framing scalable node classification methods as examples of non-parametric diffusion and highlighting their similarities; 2) Introducing Graph Non-Parametric Diffusion (GNPD), a simple method which outperforms every model with a comparable number of parameters; 3) Highlighting that non-parametric diffusion methods can be competitive in the context of node classification on homophilic graphs.

## 2 Related Work

### 2.1 Graphs

A graph $\mathcal{G} = (\mathcal{V}, \mathcal{E})$ is a collection of nodes $v_i \in \mathcal{V}$ and edges $e_{i,j} \in \mathcal{E} \subseteq \mathcal{V} \times \mathcal{V}$ between pairs of nodes. The degree of a node is its number of neighbors. In directed graphs nodes have an in-degree and an out-degree corresponding to the edges going in and out of the node respectively. A graph with $n = |\mathcal{V}|$ nodes can be described by an Adjacency Matrix $A \in \mathbb{R}^{n \times n}$ with $A_{i,j} \in \{0,1\}$ where $A_{i,j} = 1$ if $e_{i,j} \in E$ and 0 else. It is important to note that there is no canonical ordering of the nodes in a graph but they are given an arbitrary order to create an adjacency matrix. The same ordering of nodes is used to create a feature matrix $\mathbf{X} \in \mathbb{R}^{n \times d}$ with a vector of dimension $d$ representing the features of each node. In ML tasks, self-loops connecting each node to itself are often added $\tilde{\mathbf{A}} = \mathbf{A} + \mathbf{I}_N$ and the Degree Matrix $\mathbf{D}$ with $\mathbf{D}_i = \Sigma_j \tilde{\mathbf{A}}_{i,j}$ is often used to create the normalized adjacency matrix $\mathbf{S} = \mathbf{D}^{-1/2} \tilde{\mathbf{A}} \mathbf{D}^{-1/2}$.

### 2.2 Graph Neural Networks

Intuitively, GNNs aggregate information from local neighborhoods to produce more accurate node property predictions. This is done by multiplying the messages $\mathbf{M} \in \mathbb{R}^{n \times m}$ of size m, to be sent from each node by a diffusion operator. If this operator is the normalized adjacency matrix $\mathbf{S}$, each node will receive a weighted average of the messages from neighboring nodes. This process can be viewed as an aggregation with each node receiving messages from it's neighbours, or a diffusion process with each node diffusing its message to the rest of the network. By multiplying the message vectors by the $k^{th}$ power of the normalized adjacency matrix $\mathbf{M}' = S^k \mathbf{M}$, the message from each node will reach any other node within $k$ connections or less, this is called the k-hop neighborhood of a node. GNNs split this diffusion process into $k$ GNN layers separated by non-linearities. A $k$ layer GNN is then able to aggregate information from it's k-hop neighborhood. In

practice, GNNs face diminishing returns from additional layers once the number of layers goes past 3 or 4 (Kipf & Welling, 2016). Beyond the reduced benefit of adding more layers, stacking them is also expensive in terms of computation and memory which is why network depth has not been as successful for GNNs as it has in traditional deep learning methods.

The main distinction between different GNN architectures is the kind of message each node sends to its neighbors. The most general approach is to design a Message Passing GNN that can learn the most relevant messages to send between two nodes given their features. In practice, however, learning message vectors can be expensive in terms of memory (Bronstein et al., 2021), so the message sent by each node can be restricted to a weighted copy of the features of the node. The weight of these restricted messages can be learned by gradient descent as in GAT, or fixed by some heuristic as in GCN.

GCNs use the node degrees $d_u$ and $d_v$ to determine the importance of node $v$ for node $u$, with $c_{uv} = \frac{1}{\sqrt{d_u d_v}}$. While each layer of a GCN is non parametric, the update function between layers is learned. This means that for GCNs of more than one layer, the diffusion process has to be repeated for every training step. The update function of a GCN layer can be written in vectorized form as in Equation 1, for $\mathbf{H}^0 = \mathbf{X}$ and $\mathbf{W}^l$ a learnable, layer specific weight matrix.

$$\mathbf{H}^{(l+1)} = \sigma(\mathbf{SH}^{(l)}\mathbf{W}^l) \tag{1}$$

For a GCN with $k$ layers, the predicted labels are the softmax of the output from the last layer as denoted in Equation 2.

$$\hat{\mathbf{Y}}_{GCN} = softmax(\mathbf{SH}^{(k)}\mathbf{W}^k) \tag{2}$$

### 2.2.1 Inductive Bias In GNNs

An inductive bias allows us to encode assumptions into our models which can be useful to generalize to unseen data (Battaglia et al., 2018). Using the graph structure to change the representations of the nodes in order to get better predictions can be viewed as the inductive bias underpinning GNNs. The GNNs explored in this section add an increasingly strong inductive bias to the Message Passing framework. Attentional GNNs assume that each node should send a weighted copy of it's current features as a message and convolutional GNNs use a heuristic to decide these attention weights. Choosing $c_{uv} = \frac{1}{\sqrt{d_u d_v}}$ as in Kipf & Welling (2016) presents an inductive bias which gives more importance to messages coming from nodes with fewer neighbors, while other choices for the value of $c_{uv}$ result in other kinds of inductive bias. Different choices of diffusion operators like triangle based adjacency matrix (Rossi et al., 2020) can also bias the predictions in different directions.

### 2.3 Non-parametric Diffusion

For all the methods described in subsection 2.2, some parts of the aggregation process must be learned through gradient descent with back-propagation. This means the aggregation process is repeated for every training step, leading to slower training and the need for graph sampling when the graph does not fit into memory. To avoid this, Wu et al. (2019) proposed Simplified GCN (SGCN), where the non-linearities in the aggregation function are removed. This allows us to collapse the learnable matrices $\mathbf{W}^{(0)}, \mathbf{W}^{(1)}, ..., \mathbf{W}^{(k)}$ into a single matrix $\mathbf{W}$. So the prediction of a SGCN which aggregates information from its k-hop neighborhood becomes:

$$\hat{\mathbf{Y}}_{SGCN} = softmax(\mathbf{SS}...\mathbf{SX}\mathbf{W}^{(0)}\mathbf{W}^{(1)}...\mathbf{W}^{(k)}) \tag{3}$$

$$\Leftrightarrow \hat{\mathbf{Y}}_{SGCN} = softmax(\mathbf{S}^k\mathbf{X}\mathbf{W}) \tag{4}$$

Here, the matrix $\mathbf{S}^k\mathbf{X}$ can be pre-computed, so the classifier becomes a simple logistic regression model which can ignore the graph structure. This means we can use minibatches to parallelize the training process

without any sampling issues. Figure 1 depicts the contrast between this approach and that of traditional GNNs.

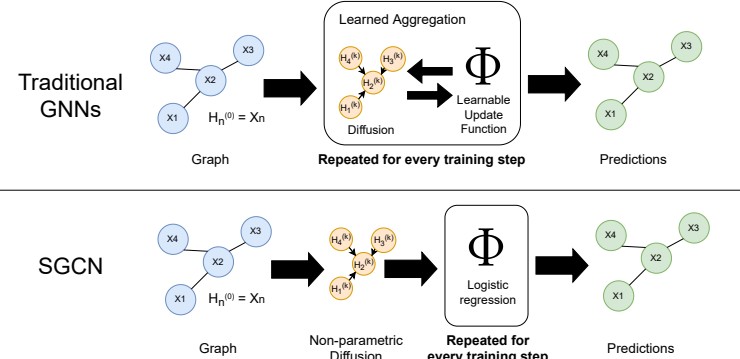

Figure 1: Schematic overview of traditional GNN methods and SGCN. In both cases the part of the method that is repeated for each training step is highlighted by a box. SGCN omits propagation from this highlighted section, resulting in classical, graph-agnostic ML classification.

Diffusion through the graph can be identified as the bottleneck of most GNN methods. Therefore, making the diffusion process non parametric to avoid having to repeat it during training has been at the core of recent attempts to make graph methods scalable (Wu et al., 2019)(Rossi et al., 2020)(Huang et al., 2020). While SGCN managed to achieve a comparable performance to GCN, both methods have fallen behind the state of the art on most benchmarks. However, newer non parametric methods like C&S, SIGN, and their variants (SAGN (Sun & Wu, 2021)) have outperformed the top learnable diffusion methods like GAT, and currently hold top spots on node classification leaderboards (Hu et al., 2020).

SIGN uses the same idea as SGCN and eliminates the non-linearities from the diffusion process, making it non parametric. Moreover, SIGN adds expressivity to its method by performing diffusion with different diffusion operators and feeding a sequence of powers of these operators through different models. The outputs of these models are then aggregated in a final dense layer that makes the final prediction.

Correct & Smooth takes a different approach to non-parametric diffusion, instead of using it as a pre-processing step to create features for a traditional classifier, it takes the predictions from a traditional classifier and performs diffusion through the graph as a post processing step. A schematic overview of both SIGN and C&S is shown in Figure 2. These methods will be explored in more detail in subsections 2.4 and 2.5.

### 2.4 Correct & Smooth

Correct & Smooth (C&S) (Huang et al., 2020) uses a graph agnostic classifier, like an MLP or a linear model to make initial prediction based on the features of each node. Assuming that the classifier makes similar mistakes on connected nodes, it uses residual propagation to address this. It further assumes that connected nodes are more likely to have the same labels and uses label propagation to enforce this. This method outperforms many GNNs with a fraction of the parameters. While C&S is accurate and fast, spectral embeddings of the graph are required as feature augmentation to achieve optimal performance.

Spectral embeddings are designed to be similar for connected nodes, therefore partially encoding the structure of the graph. They can be obtained by performing an eigendecomposition of the Laplacian of the graph, which can be computationally prohibitive for large graphs. Even with the approximation methods described in Kipf & Welling (2016), calculating the spectral embeddings is still the computational bottleneck of the C&S method.

The residuals in the correction step are scaled by a coefficient $s$ which can have a significant impact on performance for some datasets. Huang et al. (2020) propose two scaling coefficients with the choice between the two as a hyperparameter: autoscale and fixed diffusion scaling. Autoscale scales the diffused errors to

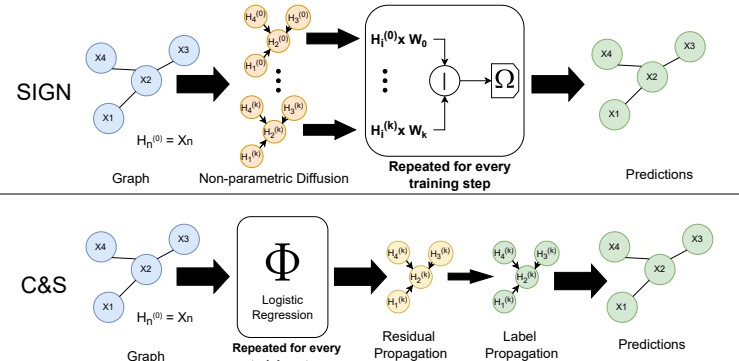

Figure 2: Schematic overview of SIGN and C&S. In both cases the part of the method that is repeated for each training step is highlighted by a box. In the case of SIGN, a non-parametric diffusion is performed before learning the matrices $\mathbf{W}_0$ to $\mathbf{W}_k$ and the classifier $\mathbf{\Omega}$. C&S trains a graph agnostic classifier and only then sends information across the graph, but the graph is still not used during training.

the magnitude of the average residuals in the training set. Fixed diffusion scaling fixes the residuals of the training nodes so they are not updated during diffusion. Fixed scale diffusion leads to label leakage which is a problem if one wants to perform learnable downstream tasks after label propagation, this will be elaborated on in Section 3.

## 2.5 SIGN

SIGN can also be seen as a set of SGCNs with different operators and receptive fields, and an additional dense layer to aggregate the outputs of all the SGCNs. The diffusion operators can be the normalized adjacency matrix $\mathbf{S}$, as in other methods, but Rossi et al. (2020) propose additional operators based on personalized page rank or a triangle based adjacency matrix. The set of linear diffusion operators is given by $\mathbf{A}^{(l)}$ for $l \in \{1, .., k\}$. The set of predictions from each SGCN is described in Equation 5.

$$\mathbf{P}^{(l)} = \mathbf{A}^{(l)}\mathbf{X}\mathbf{W}^{(l)} \tag{5}$$

With $W^{(l)}$ the learnable weight matrix, specific to each operator. A final dense layer $\mathbf{\Omega}$ is then learned to aggregate the outputs of the different SGCN layers after applying a non-linearity $\sigma$ and the concatenation operator "|".

$$\hat{\mathbf{Y}}_{SIGN} = softmax(\sigma(\mathbf{P}^{(1)}|...|\mathbf{P}^{(k)})\mathbf{\Omega}) \tag{6}$$

## 3 Methodology

Building upon the aforementioned frameworks, we propose GNPD, a novel non-parametric diffusion model for learning on graphs. GNPD employs residuals and predictions akin to C&S with an aggregation model that combines inputs from the provided diffusion operators, similar to SIGN.

### 3.1 Our Approach

In designing our approach we aim to address the following limitations of existing non-parametric diffusion methods:

1. SGCN is scalable but has fallen behind other models in terms of accuracy (Hu et al., 2020).

2. C&S relies heavily on feature augmentation with spectral embeddings which are computationally expensive for large graphs.

3. SIGN trains a model for every power of each diffusion operator, resulting in millions of learnable parameters for most graphs (Hu et al., 2020). It's accuracy has also fallen behind that of other methods.

To address these issues, we perform non-parametric diffusion in the prediction space, as in C&S, but use an aggregation model to combine different diffusion processes as in SIGN. By working in the prediction space we are able to encode prior knowledge in our diffusion process through Diffusion Bias. The result is a non-parametric diffusion method which outperforms SIGN, C&S and SGCN without the need for spectral embeddings and with orders of magnitude fewer parameters than SIGN.

## 3.2 Architecture

GNPD consists of 4 steps described in this section and depicted in Figure 3. An initial prediction is made, then used to perform residual and prediction diffusion. The outputs of the previous steps are then aggregated by a final model and smoothed with the true labels from the training data. In this section we explain each of these steps in detail.

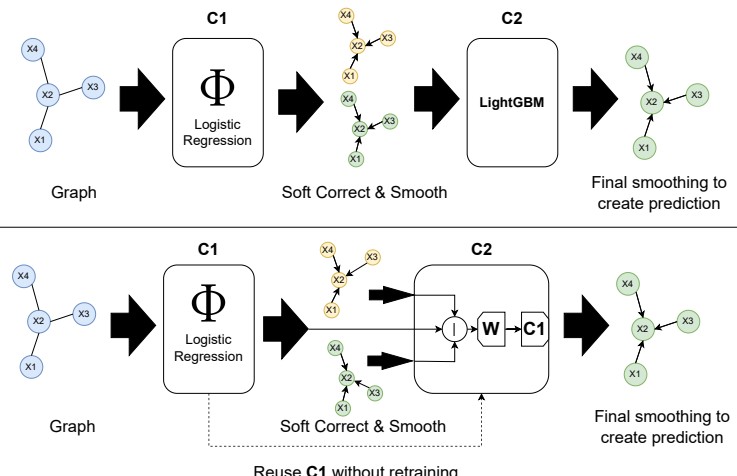

Figure 3: Schematic overview of the two versions of our method. Both present two classifiers which are trained independently, with a non-parametric diffusion of residuals and predictions in between the two. The parts of the each method that are repeated during training are shown in boxes. The symbol "|" represents the concatenation operator and $\mathbf{W}$ is a learnable weight matrix.

### 3.2.1 Graph Agnostic Classifier

An initial prediction is made by a simple graph agnostic classifier using only the features of each node. In this paper we us a logistic regression model to make these predictions. Contrary to C&S, this model does not use additional spectral embeddings. The predictions are therefore given by $\mathbf{P} = softmax(\mathbf{XW})$ using the previously established notation.

### 3.2.2 Soft Non-parametric Diffusion

The predictions from this linear model are then corrected and smoothed using our proposed Soft Correct & Smooth (SoftC&S). This version of C&S propagates predictions in the smoothing step, instead of propagating the real labels. This allows us to train a downstream classifier, as the training loss no longer goes to zero in the

smoothing process. Additionally, we bring C&S closer to other GNN methods by reducing its receptive field. The original C&S method uses 100 diffusion steps in total, resulting in each node potentially aggregating information from 100 hops away. We found that this is only viable if the spectral embeddings are included. Excluding these, a receptive field of 3 to 6 hops seems to be more adequate. This is in line with the findings of Kipf & Welling (2016) and Wu et al. (2019) which reported similar optimal receptive fields.

While the label smoothing can be replaced by prediction smoothing to avoid using the labels, the residuals cannot be computed without accessing the true labels. This is not problematic when using autoscale, since the residuals of the training nodes are also updated at every step without preserving their values from the previous step. This means the diffused residuals in the training set are representative of the residuals on the unlabeled nodes as they both arise from the same diffusion process. However, as fixed diffusion scaling treats training nodes differently from the rest, patterns found in the training data will no longer generalize to the validation and test sets. This makes fixed scaling incompatible with our method in its current form.

SoftC&S can be written as a correction step shown in Equation 8 followed by a smoothing step shown in Equation 9.

$$\mathbf{E} = \mathbf{Y} - softmax(\mathbf{X}\mathbf{W}_p) \tag{7}$$

$$\hat{\mathbf{Y}}_{Correct} = softmax(\mathbf{X}\mathbf{W}_p) + s\mathbf{S}^k\mathbf{E} \tag{8}$$

$$\hat{\mathbf{Y}}_{Smooth} = \mathbf{S}^k\hat{\mathbf{Y}}_{Correct} \tag{9}$$

### 3.2.3 Aggregation

We hypothesized that, while C&S fixes errors at a local level in the graph, this process must also be introducing errors, which could be addressed by looking at the dataset as a whole. We propose using an additional classifier after SoftC&S which will be able to address the global patterns in the errors introduced by C&S. This final aggregation can be performed by any traditional classifier, in this work we propose two approaches which are depicted in Figure 3. The first is to add a gradient boosted model (LightGBM (Ke et al., 2017)) to make the final predictions, this allows us to capture complex patterns while enabling parallelization during training. We also propose a lightweight version of our method which reuses the initial text based linear model without retraining it. We change its input with another linear model which takes the output of the diffusion steps as input. These two approaches are shown in Figure 3.

We design the aggregation classifier $\mathbf{\Omega} : \mathbb{R}^{(3c+2)\times n} \to \mathbb{R}^{c\times n}$ to take in a concatenation of the original prediction $\mathbf{P}$, the prediction after the correct step and the prediction after smoothing, along with node features like the in-degree $d_{in}$ and out-degree $d_{out}$ of the nodes. This results in the feature matrix $\mathbf{H}$ shown in Equation 10 and an aggregator prediction depicted in Equation 11.

$$\mathbf{H} = (\mathbf{P}|\hat{\mathbf{Y}}_{Correct}|\hat{\mathbf{Y}}_{Smooth}|d_{in}|d_{out}) \tag{10}$$

$$\hat{\mathbf{Y}}_{Aggregator} = \mathbf{\Omega}(\mathbf{H}) \tag{11}$$

For the second version of our method shown in Figure 3, we reuse the initial classifier $\mathbf{\Phi}$ along with the initial node features $\mathbf{X}$. The predictions from this version of our method are described in Equation 12.

$$\hat{\mathbf{Y}}_{Aggregator} = \mathbf{\Phi}(\mathbf{\Omega}(\mathbf{H}) + \mathbf{X}) \tag{12}$$

### 3.2.4 Final Smoothing

Finally we perform 2 additional smoothing steps, this time including the real labels for the training set, as no further classifiers need to be trained on this data. For the data from the validation and test sets, the true labels are not available so the predictions $\mathbf{P}$ are used.

### 3.3 Diffusion Bias

An advantage of performing diffusion in the prediction space is that predictions can be easily interpreted as the probabilities assigned by the model of each node belonging to each class. Each node is represented by a probability distribution and aggregates information from the classes assigned to it's neighbors. The interpretability of these representations allows us to introduce our background knowledge into the diffusion process in what we call Diffusion Bias. We do this in one specific way which is relevant for most real world applications, but this can be extended by using dataset specific insights.

#### 3.3.1 Class Specific Homophily

We observe that, for most graphs, homophily is not homogeneous across classes. For example, in graph citation networks, some paper topics can lend themselves to citations from a diverse set of research areas, while other research areas predominantly cite papers with the same topic. To reflect this, we add a class specific homophily coefficient based on the External-Internal (EI) homophily index (Coleman, 1964). This index looks at Internal edges which connect nodes of the same class, and External edges which connect nodes of different classes, the formula for the index is shown in Equation 13. The effect of adding class specific homophily is explored in Section 6.

$$EI = \frac{External - Internal}{External + Internal} \tag{13}$$

We use the training set to calculate the $EI_c$ index of each class $c$, an index of -1 corresponds to complete homophily while an index of 1 corresponds to the opposite (heterophily). We then set the importance of messages to be proportional to $1 - \lambda EI_c$, with $\lambda \in [0, 1]$ as a hyperparameter. As described in Subsection 2.2.1, GCNs and SGCNs use a heuristic to set the importance c of node $v$ for node $u$, $c_{uv} = \frac{1}{\sqrt{d_u d_v}}$. Class specific homophily modifies this heuristic with a different inductive bias using $c_{uv} = \frac{1 - \lambda EI_c}{\sqrt{d_u d_v}}$ instead.

### 3.4 Relationship to SGCN

We argue that the smoothing used in our method and in C&S (Equation 15) has common features with SGCN (Equation 14). The main difference between the two is whether the weight matrix $\mathbf{W}$ is learned before or after the diffusion process. SGCN aggregates node features and then makes a prediction with a logistic regression model whereas our model, like Linear + C&S aggregates the predictions of a logistic regression model.

$$\hat{\mathbf{Y}}_{SGCN} = softmax(\mathbf{S}^k \mathbf{X} \mathbf{W}) \tag{14}$$

$$\hat{\mathbf{Y}}_{Smooth} = \mathbf{S}^k softmax(\mathbf{X} \mathbf{W}_p) \tag{15}$$

### 3.5 Relationship to SIGN

A SIGN model with $k$ distinct operators also requires $k$ models like the one described in Equation 2 with $k$ distinct learnable $\mathbf{W}$ matrices. By using the kind of smoothing described in Equation 15, the $\mathbf{W}_p$ matrix is learned before diffusion in GNPD, meaning the same matrix can be used for every diffusion operator. This allows our model to combine different kinds of diffusion operators without increasing the number of parameters. In our current method, we perform two kinds of diffusion, prediction and residual diffusion. However, our approach could be extended to include other diffusion operators, like the ones used in SIGN (Rossi et al., 2020), while keeping the same amount of learnable parameters.

### 3.6 Relationship to C&S

Our method uses the same kind of residual and prediction diffusion as C&S, with some significant changes. By smoothing predictions instead of the true labels, we stop label leakage and allow for training of a

downstream model. This allows us to treat correction and smoothing as different diffusion operators for which we concatenate the outputs as input for an aggregator which makes the final prediction. Once this final prediction is made, we can make use of the training labels in a last smoothing step. Our method is also different in the lack of spectral embeddings and the inclusion of class specific homophily.

## 4 Experimental Setup

### 4.1 Datasets

| Dataset | Nodes | Edges | Classes | Train/Val/Test |
|---------|-------|-------|---------|----------------|
| arxiv | 169,343 | 1,166,243 | 40 | 54%/18%/28% |
| Products | 2,449,029 | 61,859,140 | 47 | 10%/2%/88% |
| Pubmed | 19,717 | 44,338 | 3 | 92%/3%/5% |
| Citeseer | 3,327 | 4,732 | 6 | 55%/15%/30% |

Table 1: Summary statistics for the datasets used in this paper.

The statistics of the datasets used in this paper are described in Table 1.

### 4.2 Additional Text Features

As described in Subsection 4.1, most graph benchmarking datasets come with text embeddings to represent the node features instead of using the raw text corresponding to each node. This equalizes the information that each method can extract from the text and puts the focus on learning from the graph structure. However, recent methods (Chien et al., 2021) have started making use of additional text features to learn better node representations, outperforming any models that do not use these additional features (Hu et al., 2020). In this work we evaluate our models on all datasets in the originally intended setting, without additional text features. However, we also use the text of the papers to generate better text encodings for ogbn-arxiv using a sentence transformer (Reimers & Gurevych, 2019). This allows us to evaluate how much our method can benefit from additional text features and how it compares to state of the art ensemble models based on GIANT (Chien et al., 2021) which are designed to make use of these additional features.

## 5 Results

Performance metrics and model parameter counts for experiments on ogbn-arxiv without additional text features are denoted in Table 2. Our method is compared to the SOTA and the architectures it draws inspiration from, SIGN and C&S. Notably, our method outperforms SIGN, all iterations of C&S and achieves close to SOTA performance with only 1% of the parameters. This parameter efficiency is highlighted in Figure 4a.

Table 3 compares GNDP performance on ogbn-arxiv using additional text features to that of the overall SOTA ensemble using this feature augmentation. This inclusion boosts our test accuracy to $74.76 \pm 0.03\%$. Figure 4b depicts the contrast in parameter efficiency between GNDP and the overall best performing architectures on ogbn-arxiv.

In Table 4 we evaluate our method on three additional datasets. We see improvement over our baselines in Pubmed and Citeseer, but not on ogbn-products. While our method outperforms the autoscale version of C&S along with the other baselines, it does not outperform C&S with Fixed Diffusion (FixedDiff) Scaling on ogbn-products. FixedDiff makes a minor difference on every dataset reported in Huang et al. (2020) except for ogbn-products where it leads to an increase in accuracy of 5.58%. Since our method is incompatible with FixedDiff Scaling, it cannot benefit from this boost in performance and therefore only outperforms the autoscale version of C&S on ogbn-products.

| Method | Validation Accuracy | Test Accuracy | Parameters |
|---|---|---|---|
| MLP + C&S | 72.42% ±0.06 | 71.04% ±0.09 | 85,323 |
| Linear + C&S | 73.00% ±0.01 | 71.18% ±0.01 | 15,400 |
| GCN | 73.00% ±0.17 | 71.74% ±0.29 | 110,120 |
| SIGN | 73.23 ± 0.06 | 71.95% ±0.11 | 3,566,128 |
| Linear + C&S + embeddings | 73.68% ±0.04 | 72.22% ±0.02 | 15,400 |
| MLP + C&S + embeddings | 73.91% ±0.15 | 73.12 ±0.12 | 175,656 |
| GNPD (Linear) | 74.49% ±0.06 | 73.51% ±0.04 | **15,400** |
| GNPD (LightGBM) | 74.68 ±0.08 | 73.72% ±0.04% | _ |
| AGDN+BoT+self-KD+C&S | **75.19** ±0.09 | **74.31** ±0.14 | 1,513,294 |

Table 2: Validation and test metrics on ogbn-arxiv. Reported accuracy scores are averages over 10 runs along with the corresponding standard deviations. Best performing results are depicted in **bold**, while the best results out of our baselines are underlined.

| Method | Additional text | Validation Accuracy | Test Accuracy | Parameters |
|---|---|---|---|---|
| GIANT-XRT+DRGAT+KD | Yes | **77.25%** ±0.08 | **76.33** ±0.06 | 2,685,527 |
| GNDP (Linear) | Yes | 76.20% ±0.04 | 74.76% ±0.03 | **15,400** |
| GIANT-XRT+GraphSAGE | Yes | 75.95% ±0.11 | 74.35% ±0.14 | 546,344 |
| AGDN+BoT+self-KD+C&S | No | 75.19 ±0.09 | 74.31 ±0.14 | 1,513,294 |

Table 3: Accuracies over 10 runs and number of parameters on ogbn-arxiv using additional text features. We also include the SOTA model without text features to highlight the importance of better node features over complex aggregation methods in homophilic graphs.

For ogbn-products, the SOTA model listed in Table 4 corresponds to GAMLP+RLU+SCR+C&S, an ensemble model with 3,335,831 parameters. For Citeseer and Pubmed, the SOTA models are currently versions of Snowball (Luan et al., 2021).

With respect to model runtimes, avoiding the use of spectral embeddings results in a substantial increase in efficiency, with our model scaling better to larger datasets than methods using spectral embeddings, like C&S. The benefits of spectral embedding removal due to increasing computational cost for larger datasets is highlighted in Figure 5a.

## 6  Analysis

Our results show that on common benchmarking datasets, complex diffusion processes with millions of learned parameters are not needed to achieve a competitive performance. In fact, we achieve this without any learnable parameters in the diffusion process. This suggests that in practice, for many tasks in the real world, edges encode similarity, making smoothing in the prediction space very effective, even without learnable weights. We show that in this setting, pairing a simple method like GNPD with improved node features has a bigger impact on accuracy than using complex models to find patterns in the graph (Table 3). Even without additional text features, our method competes with models that have orders of magnitude more parameters. By making diffusion non-parametirc, the number of parameters in our method does not depend on the size of the graph, but only on the size of the node features.

### 6.1  Ablation study

In this section we measure the importance of the different components of our method. While most of the hyper-parameters in this method were inherited from C&S, we introduced the $\lambda$ parameter to regulate

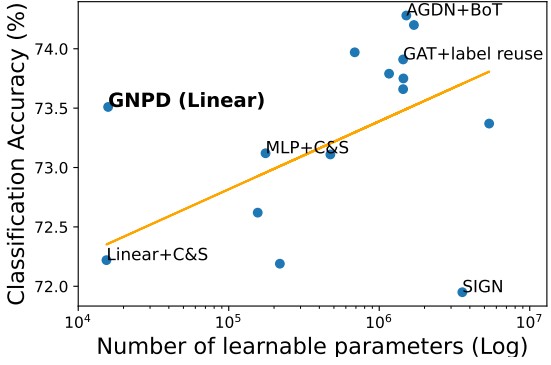

(a) Parameter efficiency of top methods without additional text features.

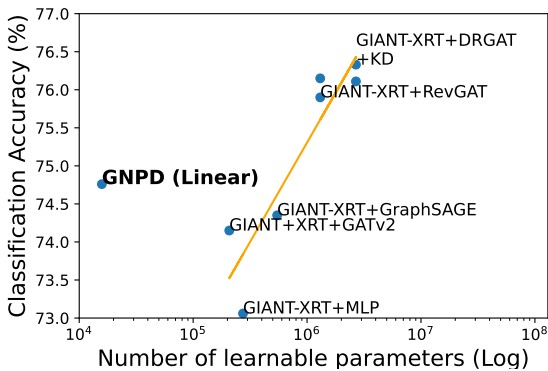

(b) Parameter efficiency of top methods using additional text features.

Figure 4: Parameter efficiency of the top methods on the ogbn-arxiv leaderboard with and without using additional text features. A regression line is added to show the exponential relationship (linear in log scale) between parameters, and accuracy.

| Method | Pubmed | Citeseer | ogbn-products |
|---|---|---|---|
| SGCN | 84.04 | 72.04 | — |
| GCN | 88.13 | 73.68 | 75.64 |
| Linear+C&S autoscale | 89.99% | 76.31% | 80.20% |
| Linear+C&S FixedDiff | 89.74% | 76.22% | 82.54% |
| MLP+C&S+embeddings autoscale | 89.33% | 76.31% | 78.60% |
| MLP+C&S+embeddings FixedDiff | 89.23% | 76.42% | 84.18% |
| GNPD (Linear) autoscale | 90.02% ±0.04 | 79.32% ±0.21 | 82.80% ±0.08 |
| SOTA | **91.44% ±0.59** | **82.07% ±1.04** | **85.20% ±0.08** |

Table 4: Test accuracies averaged over 10 runs with the corresponding standard deviation when available. SOTA results are shown in **bold**, while the best results out of our baselines are underlined.

class specific homophily; we show the sensitivity of our method to this hyper-parameter and the number of diffusion steps $k$ in Table 5.

Our method can be split in four parts, described in Section 3, we show how much each of these parts contribute to the final accuracy in Table 6. We see non-parametric diffusion drastically increases the accuracy of the initial Linear model, from 52.32% to over 70%, which higlights the homophily of the graph. Interestingly, we see accuracy goes down after the aggregation step. We hypothesize that this is due to the graph agnostic nature of the aggregator which makes it unable to fully preserve the smoothing of the previous steps. Therefore, the improvements made by the aggregator are initially outweighed by giving up on the smoothness of the prediction. However, after reintroducing this smoothness with the final smoothing with labels, we get an accuracy of 73.50%. By comparison, if we skip the aggregator and go directly to the final smoothing with labels, the final accuracy drops to 69.09%. This happens because the output of the "Correction and Smoothing" step is already smooth, and smoothing it further leads to oversmoothing, while the output of the aggregator is not smooth and therefore does benefit from smoothing.

## 6.2 Limitations

While GNPD is a very simple model with unmatched parameter efficiency, the implications of this for scalability and speed are not thoroughly explored in this paper. We show that, by foregoing spectral embeddings,

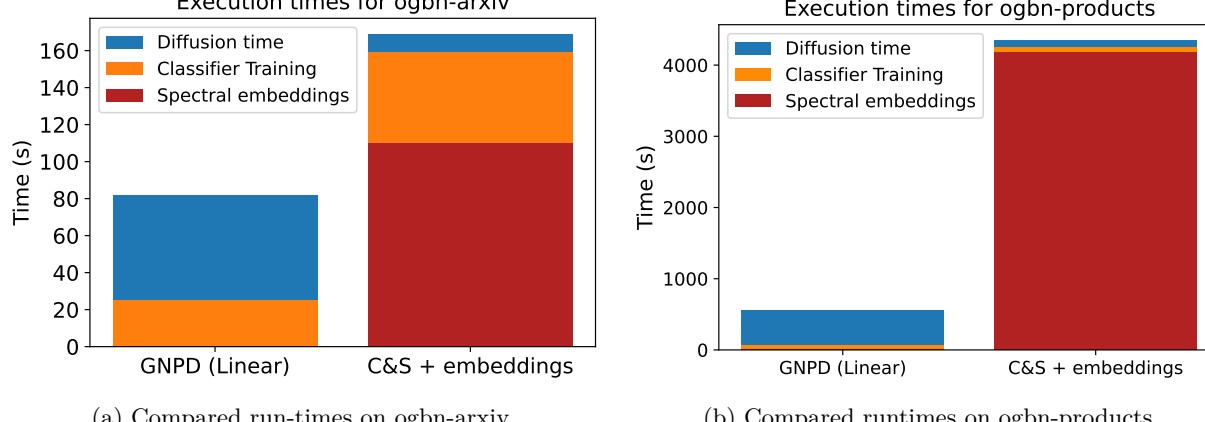

(a) Compared run-times on ogbn-arxiv.

(b) Compared runtimes on ogbn-products

Figure 5: Compared runtimes of GNPD (Linear) and the closest version of C&S in terms of accuracy (MLP+C&S+embeddings), averaged over 10 runs on 2 datasets. We see the spectral embeddings are the most computationally expensive part of C&S and that by removing them, our model divides the training time by 2 in the case of ogbn-arxiv and by 10 in the case of ogbn-products.

| Method | $\lambda$ | Test Accuracy on arxiv | k | Test Accuracy on arxiv |
|---|---|---|---|---|
| C&S +Linear | - | 71.26 ±0.01 | 50 | 71.26 ±0.01 |
| C&S + Linear + embeddings | - | 72.22 ±0.02 | 50 | 72.22 ±0.02 |
| | 0 | 72.65% ±0.08 | 0 | 51.95 ±0.6 |
| | 0.1 | 72.87% ±0.7 | 1 | 72.52 ±0.7 |
| | 0.2 | 73.19% ±0.08 | 2 | 72.13 ±0.7 |
| GNDP (Linear) | 0.3 | 73.46% ±0.06 | 3 | **73.50** ±0.7 |
| | 0.4 | **73.50%** ±0.07 | 4 | 72.98 ±0.8 |
| | 0.5 | 73.46% ±0.14 | 5 | 73.20 ±0.6 |
| | 0.6 | 73.32% ±0.08 | 6 | 73.02 ±0.7 |

Table 5: Sensitivity analysis with respect to $\lambda$ and k (the number of diffusion steps). The results are averaged over 10 runs and shown with standard deviation.

our model is substantially faster than C&S, particularly on large graphs. However, a more comprehensive comparison to the run-times, time and memory complexities of other methods would be required to make confident claims about the scalability of this method. We also want to emphasize that this method is only tested and expected to work in homophilic graphs.

## 7    Conclusion

We introduced GNPD, a novel method for non-parametric diffusion, designed to enable parameter efficient node classification while improving accuracy over existing non-parametric diffusion models. In doing this, we grounded the C&S method in the same theoretical framework as other GNNs, allowing us to draw from insights in the GNN literature to design our method and forego the computational bottleneck of spectral embeddings. Our work shows that graph agnostic linear models can be alternated with non-parametric diffusion to achieve competitive results with unmatched parameter efficiency on homophilic graphs. Further work could explore the possibility of adding different diffusion operators to our method as in SIGN (Rossi et al., 2020), since our method makes it possible to do this without multiplying the number of learnable parameters.

| | Test Accuracy on arxiv | | |
| --- | --- | --- | --- |
| | GNPD | Linear+C&S | Linear+C&S+embeddings |
| Linear model output | 52.32% | 52.32% | 52.32% |
| Correction and Smoothing | 71.5% | 71.26% | 72.22% |
| Aggregation output | 69.74% | - | - |
| Final Smoothing with labels | **73.50**% | - | - |

Table 6: Ablation study of the different parts of our method compared to C&S + Linear with and without embeddings. Note that the Smoothing step uses predictions in the case of GNPD and labels in the case of C&S.

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
