# OpenReview forum: "Parameter Efficient Node Classification on Homophilic Graphs"
_TMLR — Accepted by TMLR_

### Review · Reviewer_MXvx · 2023-01-03

**Summary Of Contributions:**

This paper studies an interesting problem, i.e., using non-parametric diffusion in graph node classification tasks, to make the model scalable.

The proposed model shows better accuracy performance than SGCN and less computation-consuming than SIGN and C&S.

**Audience:**

Yes

**Claims And Evidence:**

No

**Requested Changes:**

1. Experiments
	1. Experiments on larger graph datasets are expected to show the scalability.
	2. The computation complexity/memory consumption/running time are expected. (Only comparing the running time with C&S is insufficient)
2. Methods
	1. More theoretical analysis or insights of the proposed method are expected, e.g., why the non-parametric methods can achieve comparable performance.


**Strengths And Weaknesses:**

### strengths

1. The task to use low amount of parameters in node classification models is interesting.
2. The proposed method mitigates the high cost of spectral embedding in C&S.

### weaknesses

1. The experiments of the proposed **scalable** node classification method on large-scale graph datasets are insufficient.
	1. There are some large graph datasets, e.g., ogbn-papers100m, ogbn-mag240m can be considered to show the scalability of the proposed method
	2. Though the number of model parameters is reported, the time complexity, running time, memory consumption are also expected to support the claim of **scalable**.
2. The design of combination of C&S, SIGN, SGCN seems quite ad-hoc and straighforward. More theoretical analysis or insights of the proposed method are expected, e.g., why the non-parametric methods can achieve comparable performance.

---

> ### Author Response · Authors · 2023-02-08
> **Reply**
>
> Thank you for the constructive review, we will address your comments and requested changes below:
>
> 1) To support the claim of scalability, we will evaluate our method on ogbn-papers100m and discuss the time complexity of these methods.
>
> 2) Part of the reason why non-parametric methods can be successful is the homophilic nature of node classification datasets. We will highlight this and provide further discussion on this topic.

---

> > ### Author Response · Authors · 2023-02-16
> > **-**
> >
> > We thank the reviewer again for taking the time to review our paper. Given the time constraints and our hardware limitations we have chosen to adjust our claims to better align with our evidence and removed the claim of scalability, focusing on parameter efficiency instead. We agree that a more thorough analysis of the time complexities of the different methods and an evaluation on one of the largest datasets are required to fully establish the scalability of this method. However, we believe that we have enough evidence of the accuracy and parameter efficiency of our model to make this a paper of interest for some individuals in TMLR's audience.
> >
> > Evaluating our method on larger datasets would be an interesting avenue for future work since, by contrast to most GNN models, the parameters of our method do not scale with the size of the graph.

---

### Review · Reviewer_RaZ3 · 2023-01-26

**Summary Of Contributions:**

The authors propose a new aggregation scheme aimed towards decreasing number of learnable parameters in GNNs. Towards this they propose a framework which roughly works in four stages: (1) Initial prediction using a logistic regression model (2) Label correction and smoothing using averaging over neighbors (3) Re-prediction using smoothened predictions using another model (4) Final smoothing using neighborhood averaging. Because smoothing steps are decoupled from weights learning steps, number of learnable parameters doesn't depend on the diameter of averaging and thus results in significantly fewer number of parameters compared to traditional GNNs.
The proposed method achieves close to SOTA results on 4 datasets.

**Audience:**

Yes

**Broader Impact Concerns:**

None.

**Claims And Evidence:**

Yes

**Requested Changes:**

(1) The authors should conduct ablation studies to measure impact of the third step (referred to as the aggregation in 3.2.3) and final smoothing (section 3.2.4).

(2) Comparison with other relevant methods for example: C&S+ Linear  is only reported in Table 2 ogbn-arxiv, while missing from Table 4. Please include them for other datasets.

(3) If the authors can demonstrate the effectiveness of the proposed approach on other modality dataset/task (visual, audio) that will greatly strengthen the experiment section.

(4) Corresponding to figure 5, also include a table that compares total runtime of different methods.

**Strengths And Weaknesses:**

Strengths:
(1) The main idea is clearly written.
(2) Faster runtime of the proposed method compared to existing methods.
(3) Performance boost is okayish compared to C&S and SIGN.

Weaknesses:
(1) The proposed method is conceptually very similar to the Correct and Smooth(C&S) approach by Huang et al. Pretty much the only differences being the altered sequence of weight learning and smoothing procedure (proposed method also don't use spectral embedding).

(2) The results section is unstructured. For example, in table 4 the comparisons with C&S+ Linear is missing which is very relevant. See requested changes section.

(3) Effects of steps like 3.2.3 and 3.2.4 are not studied properly. See below.

---

> ### Author Response · Authors · 2023-02-08
> **Reply**
>
> Thank you for the constructive review. We will address the changes you requested below:
>
> (1) We agree that the paper would benefit from an ablation study and are currently implementing this change.
>
> (2) We will include C&S + Linear for other datasets and also C&S + Linear without spectral embeddings as it is the closest version of C&S to our method. Note that our method outperforms C&S + Linear without spectral embeddings on every dataset we tested, and in some cases by several percentage points (Citeseer, arxiv).
>
> (3) We agree with the reviewer that it would be interesting to evaluate this method on other modality datasets, however most benchmarking datasets for node classification are based on text and these are the datasets our baseline methods are evaluated on. We will therefore be unable to evaluate on other modalities of data in this time frame as we would also need to adapt our baselines to these modalities.
>
> (4) In figure 5, we compare the runtime of our method against that of the closest version of C&S in terms of accuracy. Any version of C&S that uses spectral embeddings will be slower than our method, as calculating the spectral embeddings alone takes longer than training our method. For every other method we provide the number of parameters but think the runtime comparison is less relevant. SIGN is less competitive in terms of accuracy with 2 orders of magnitude more parameters and the SOTA models we list also have 2 orders of magnitude more parameters and cannot be evaluated using the same hardware we used for our experiments due to their size.

---

### Review · Reviewer_xc7N · 2023-02-03

**Summary Of Contributions:**

Inspired by the success of C&S, SIGN, and SGC the authors propose GNDP that aims to integrate the stregths of the aforementioned approaches while avoiding their weakness. Similar to other "decoupled" GNNs, the diffusion and feature processing steps are separate from each other which allows for efficiency.

**Audience:**

Yes

**Broader Impact Concerns:**

Not applicable.

**Claims And Evidence:**

Yes

**Requested Changes:**

I propose that the authors include a detailed ablation study as well as sensitivity analysis w.r.t. the hyperparameters.

In addition I think the paper will benefit from a rewrite to reduce the amount of expository explanation. Currently a large portion is dedicated to explaining the background and the baseline methods. These sections can be written significantly more succinctly, which will help better highlight the contribution of this paper.

Overall, while I believe that the simplicity of the approach is a strength, given the lack of technical contributions, I suggest the empirical analysis to be straightened by including the above suggestions, as well as more detailed analysis of the limitations and performance of the method on different types of graphs (e.g. graphs with heterophily).

**Strengths And Weaknesses:**

- The biggest strength of the method is its parameter-efficiency which is most clearly evident in Figure 4 where GNDP can achieve satisfactory accuracy (but lower than SOTA) with orders of magnitude fewer parameters.
- The simplicity of the approach is another strength.

- A big weakness of the proposed approach is the lack of an ablation study. For example the authors introduce several additional modifications such as correcting for class-specific homophily (3.3.1) and confidence (3.3.2). It is not clear how important these additions are. Moreover, would adding these modification to the baselines also improve their performance?
- Similarly, the final classifier concatenates several different sources of information (Eq. 12) and again it is not clear which of these is crucial for obtaining good performance.
- Another weakness is the lack of a sensitivity analysis. and how sensitive is the method to the hyperaparameters, e.g. $\lambda, s, k$, etc.
- The paper lacks a discussion of the limitations of the proposed approach. For example, similar to the methods it is inspired by, its good performance is likely highly dependent on the degree of homophily.

---

> ### Author Response · Authors · 2023-02-08
> **Reply**
>
> Thank you for the constructive review, we agree that the paper would benefit from a more thorough analysis as well as an ablation study and sensitivity analysis, we will work to implement these changes. We will also work to make the scope and limitations of our method clearer by adding a more detailed analysis and highlighting the importance of homophily for the success of this method. As suggested, we will make the background section shorter and use this space to implement the proposed changes.

---

### Decision · Action_Editors · 2023-03-08

**Recommendation:** Accept with minor revision

**Comment:**

As noted in my comments on claims and evidence, the key claims of the paper, #1 and #2, are well supported.  There were concerns about claim #3 and concerns about the claim of “scalability”, which I believe could be addressed by the authors with a new revision.

All reviewers lean on the rejection side for this paper, potentially due to limited novelty.  But I found the paper hard to reject according to the TMLR acceptance criteria, as it passes both the claims and evidence threshold and the audience threshold.  Therefore I’m recommending an acceptance but request the authors to revise the paper further, in particular addressing the concerns highlighted in the claims and evidence comment.

I’d also encourage the authors to consider one reviewer’s suggestion about the limitation of the proposed approach, which would help make the paper better.

**Audience:**

Researchers interested in graph neural networks and in particular efficient and scalable graph neural networks would find this work interesting.

**Claims And Evidence:**

This paper proposes a parameter-efficient node classification model with a slight modification of the previously published “correct & smooth” method.  The parameter-efficiency is achieved by only using 2 linear layers and have the entire graph-diffusion / message passing process parameter-free.

The paper claims 3 contributions:
1. Re-framing scalable node classification as non-parametric diffusion.  This is supported by Section 2 of the paper which does a good job of reviewing previous literature and highlighting the non-parammetric diffusion aspect.
2. Introducing the Graph Non-Parametric Diffusion (GNPD) method and show it outperforms existing models.  This is supported by the experiment results and extended analysis, showing GNPD performing better than in particular the “correct & smooth” baseline.
3. Highlight the sufficiency of non-parametric diffusion for node classification on homophilic graphs.  This point is rather questionable, as GNPD alone is clearly not sufficient and there exists other SOTA methods that perform better than the proposed GNPD as reported in various results tables.  In fact it is hard to argue any method being sufficient to solve the node classification problem when the classification accuracy is still just ~75% for some tasks.

Notably the reviewers also raised concerns about claim #2, where in the first version of the paper the experiment results were significantly weaker, only reporting a few “final” performance results without analyzing the contribution from the different components introduced by the paper.  The later added analysis and ablation results strengthened the results quite a bit.

Another concern was raised about the “scalable” part of the claim and the narrative of the paper, as the experiments were not done on very large scale datasets.  The authors acknowledge that they take back the claim on the “scalability” front and instead focus on parameter efficiency.  However much of the narrative of the paper still has an emphasis on scalability.